# Parental, Teacher and Peer Effects on the Social Behaviors of Chinese Adolescents: A Structural Equation Modeling Analysis

**DOI:** 10.3390/brainsci13020191

**Published:** 2023-01-23

**Authors:** Chao Huang, Cheng Li, Fengyi Zhao, Jing Zhu, Shaokang Wang, Jin Yang, Guiju Sun

**Affiliations:** 1Key Laboratory of Environmental Medicine and Engineering of Ministry of Education, Department of Nutrition and Food Hygiene, School of Public Health, Southeast University, 87 Dingjiaqiao Road, Nanjing 210009, China; 2Institute of Biotechnology and Health, Beijing Academy of Science and Technology, Beijing 100089, China; 3Department of Public Health, School of Medicine, Xizang Minzu University, Xianyang 712082, China

**Keywords:** adolescent, behavior development, parent, teacher, peer

## Abstract

Adolescent behavior is closely related to academic and long-term personal development, and adolescents are vulnerable to the influences from people around them. This study aimed to analyze the factors and mechanisms that influence the behavior of adolescents. It examines the impact of family, teachers, and peers on adolescent prosocial behavior and misconduct. Data were obtained from the China Education Panel Survey (CEPS) follow-up data (2014–2015 school year) and 7835 middle school student participants were used for analysis. Structural equation modeling (SEM) was used to explore the influence and mechanisms of family, teachers, and peers on the development of adolescent social behavior. The findings showed that parental relationships, parental discipline, teacher supervision, and positive peer behavior were positively associated with adolescent prosocial behaviors and reduced the incidence of delinquent behaviors, while frequent home–school contact was associated with misconduct (all *p* < 0.01). These results remained significant after controlling for gender, residence, only-child status, family financial situation, and paternal education. Significant others in an adolescent’s life play multiple essential roles in forming and developing adolescent behavior and in directly influencing them. To guide the prosocial behaviors of middle school students and reduce delinquent behavior, we should build harmonious parent—child, peer, and teacher–student relationships, teach according to their aptitudes, pay attention to particular groups and strengthen psychological health education to develop their self-esteem and self-confidence.

## 1. Introduction

### 1.1. Background

The wellness and development of adolescents are increasingly valued, especially in terms of psychological and behavioral development [1]. However, young people’s management remains a challenge for society, schools, and families alike. The socio-behavioral development of adolescents affects their life trajectory and is also closely related to society’s development as a whole [2]. Numerous studies have shown that student behavioral development impacts academic performances in the short term [3,4], but also predicts educational attainment in the medium to long term [5,6]. Adolescence is a crucial period in life. Psychological and social abilities are developing rapidly but are immature and vulnerable to significant others’ influence [7]. According to the American sociologist Mills, a significant other is a person who has an essential impact on an individual’s socialization. Parents, teachers, and peers are in close contact with adolescents’ school life and play different roles, influencing adolescent behaviors and even triggering misconduct [8,9,10]. Therefore, it is necessary and relevant to investigate the influences and mechanisms of the different levels of impact on social behavior development during adolescence.

As socialization progresses, adolescents gradually adapt to their social roles, acquire social skills and exhibit different social behaviors. Scholars generally agree that social behavior can be divided into two types: prosocial behaviors and misconduct [11]. Weisberg, an American scholar, was the first to coin the term ‘prosocial behavior’, which, in his view, refers to sharing and helping behavior, the common denominator of which is the benefit to society and others. Since then, scholars have defined prosocial behavior from various perspectives, and most definitions emphasize the socially condoned nature of the action. In general, despite the cultural differences between China and the West, both Western and Chinese scholars classify prosocial behavior as a range of measures that benefit others, including specific actions such as helping, complying, being loyal, and promoting friendship [12,13]. The definition of misconduct, on the other hand, is still evolving and being refined, and definitions are still emerging. Although there is no consensus on what constitutes delinquent behavior, there are certain commonalities. As a form of negative social behavior, it refers to a range of actions that are not in line with social norms and norms in social interactions or that do not fit well into society in a way that causes adverse effects or even harm to others and themselves, including indiscipline, aggression, temper tantrums, and others [14,15].

### 1.2. Family, Teacher, Peer Effects

Early research has found that various parenting methods, such as inductive discipline, empathic didactics, positive behavior modeling, and responsibility allocation, help promote social development. Parents can have a positive and profound influence on their children’s behavioral development for three main reasons: Firstly, by investing in and guiding their children’s education, parents pass on their ideas about the importance of education, allowing their children to internalize these values and put them into practice in daily behavior [16,17,18]. Secondly, parents can establish links with their children’s teachers and peers and selectively supervise them in a relatively closed social network [19]. Thirdly, parents have the advantage of being ‘insiders’ and have access to a wide range of information to better inform and correct their children when a problem behavior occurs [17,18]. Nevertheless, some studies have also concluded that parental supervision or psychological control may be correlated with children’s delinquent behavior. Strict discipline can cause children to lose their sense of freedom and exacerbate reactive behavior [20]. Poor family relationships are associated with antisocial or problematic behavior in adolescents, for example, harsh parenting styles and high-conflict family environments undermine adolescents’ mental health and increase the risk of antisocial behavior and substance use [21,22]. As seen, the family’s role in adolescent behavior development still varies and needs further validation. Our first hypothesis was that good parent-child relationships and parental supervision facilitate the development of prosocial behavior in children and reduce the incidence of misconduct.

Adolescent students spend most of their time in school, apart from home. During this period, teachers take on the responsibility of supervising children in place of parents and significantly impacting young people’s normal development [23]. Teacher supervision is often thought of as classroom management, including teacher attention, questioning, and commendations [24]. Sanches’s research shows that fair treatment, combined with awareness from teachers, increases the student’s sense of belonging to the school, acceptance of the school system and rules, and further compliance with laws and regulations for social promotion [25]. A study among Norwegian adolescents also confirmed that more attention and support from teachers resulted in a lower incidence of aggressive behavior among students [26]. Scholars have argued that students display good prosocial behavior when teachers’ verbal encouragement promotes heightened academic performance and boosts their confidence in learning and life [24]. However, some scholars disagree with this view and believe that a high level of teacher supervision and attention does not necessarily apply to all students [27]. High-pressure, low-care supervision by teachers or one-way management that lacks interaction can have an impact on students’ academic performance and mental development. Therefore, the role of teacher supervision in adolescent behavioral development needs further validation. Our second hypothesis was that teacher supervision (which involves attention, question, and commendation), facilitates the development of prosocial behavior and reduces the incidence of delinquent behavior.

Peers are a vital interaction group in adolescent development, and their behavior can significantly influence behavioral developmental tendencies [10]. Adolescents will identify with the values of their peers and imitate their behavioral habits to improve their homogeneity and integration into the group [28]. Research has shown that adolescents who make friends with those more likely to engage in delinquent or antisocial behavior are themselves more likely to engage in these behaviors than when they befriend peers who are more prosocial [29]. Peer effects can be generated by two main pathways, direct and indirect impacts. Direct influence means that peers, as personal resources, can provide social support for a young person’s academic or personality development. The excellent academic performance or personality of peers can be contagious and directly influence the surrounding students. Peers helping each other, sharing problems and discussing concerns are all direct positive examples of the peer effect. Similarly, bad behavior such as smoking and drinking habits, as well as aggressive behavior, can be directly transmitted and reflected among peers. Indirect influence, on the other hand, places greater emphasis on the peer network as a social context, which acts indirectly by shaping individual perceptions and behavior, meaning that positive or negative peer behavior or performance does not directly affect the individual, but rather acts by influencing classroom behavior, teaching progress, class climate and the teacher’s attitude towards the class group as a whole [30]. However, whether peer behavior can act directly on adolescents themselves without indirectly influencing their behavior through other external conditions needs to be clarified [31]. The third hypothesis we made was that peers’ prosocial and deviant behavior would directly influence adolescents’ behavioral development, with peers excelling in prosocial behavior increasing the good performance of other adolescents around them and reducing deviant behavior, while peers’ deviant behavior would have the opposite effect.

### 1.3. Research Interest

The development of adolescent behavior has received increasing interest in the literature, and a large number of a priori findings have laid the foundation for future theoretical and practical products. However, there is still room left for improvement. Most studies have taken a single perspective on the influence of adolescent delinquency, with the independent and dependent variables in only one dimension. However, the human environment is complex and interactive, with parents, teachers, and peers influencing adolescent values and behaviors differently. There are still controversies and gaps in previous research, with scholars not reaching a consensus on parental and teacher supervision and insufficient research on whether parent–school contact impacts adolescent behavioral development. Whether the mechanisms of influence of significant others on adolescents in different cultural contexts remain reliable remains to be studied.

Thus, in this paper, parents, teachers, and peers were all considered exogenous variables, and the endogenous variables included both positive and negative aspects of adolescent behavioral development. Structural equation modeling was used to investigate the underlying mechanisms.

## 2. Materials and Methods

### 2.1. Data and Sample

This study used data from the China Education Panel Survey (CEPS), a large-scale nationally representative tracking survey designed and implemented by the National Survey Research Centre (NSRC) at Renmin University of China. The survey was conducted at the student, parent, teacher, and school levels to investigate adolescents’ educational performance and behavioral development. CEPS used the 2013–2014 school year as the baseline, and 28 county-level units (counties, districts, and cities) were randomly selected as survey sites from across the country using the average level of education of the population and the proportion of the nonresidential, floating population as stratification variables. The survey was implemented on a school basis, with 112 schools and 438 classes randomly selected for the study in the selected county-level units. All students in the sampled grades were enrolled, and a total of approximately 20,000 students were surveyed in the baseline survey. Two waves of the survey were conducted: a baseline survey for the 2013–2014 school year and a follow-up survey for the 2014–2015 school year. This study selected follow-up data from the 2014–2015 school year, with essential information matched to the baseline data. We further excluded participants with missing data on critical variables, resulting in a final analysis sample of 7835 participants. All of the participants were in eighth grade.

### 2.2. Measures

#### 2.2.1. Endogenous Variables

The endogenous variable in this study was the adolescents’ social behavior, which, in light of other studies in the literature, includes two main aspects: prosocial behavior and misconduct. Therefore, we used question D1 in the student questionnaire to measure pro-sociality, namely, “Whether you could do the following in the past year?” including the three observation variables of “helping the elderly” (D1-1), “obeying orders, voluntarily queuing” (D1-2) and “being sincere and friendly to others” (D1-3), which were used to define the latent variable of “pro-sociality”. Question D2 measured misconduct: “In the past year, have you had any of the following problems?” including “swearing (D2-1), quarreling (D2-2), fighting (D2-3), bullying (D2-4), feeling grumpy (D2-5), being inattentive (D2-6), truancy (D2-7), plagiarizing homework (D2-8), smoking and drinking (D2-9), going to internet cafes and game halls (D2-10)”. These ten observational variables define the latent variable “misconduct”. The questions above were scored on a five-point Likert-type scale from “never = 1” to “always = 5”, with a higher score indicating a higher frequency of occurrence.

After considering the research needs and the validation factor analysis results, the endogenous latent variables were adjusted as follows: The latent variable “pro-sociality” was defined using the two observed variables, “mannered” and “friendly”, denoted by D1-2 and D1-3, respectively. D1-1 was excluded because its factor loading was below 0.55. In addition, the latent variable “misconduct” was defined as the two observed variables, “self-behavior” and “language & psychology”. The former was expressed by the sum of questions D2-3, D2-4, D2-7, D2-8, D2-9, and D2-10, forming a variable with values in the range 6-30 (Cronbach’s alpha = 0.74). The second was represented by the sum of questions D2-1, D2-2, D2-5, and D2-6, forming a variable with values in the range of 4–20 (Cronbach’s alpha = 0.74).

#### 2.2.2. Exogenous Variables

The family effect was represented by three latent variables: parental relationship, parental discipline, and parent–school contact. The parental relationship was measured by the two observed variables, “relationship with father” and “relationship with mother”, and was scored on a three-point scale of “not close = 1, average = 2, very close = 3”. Parental discipline was assessed by question A20, “Are parents strict in the following matters?” including “homework (A20-1), school performance (A20-2), friendships (A20-3), dress (A20-4), time spent online (A20-5), and time spent watching television (A20-6)”. These six items were measured on a three-point scale, with “do not care = 1, care but not strictly = 2, care a lot = 3”. Based on previous research, these were combined into “study” (sum of A20-1 and A20-2), “life” (sum of A20-3 and A20-4), and “electronics” (sum of A20-5 and A20-6) as the three observed variables for “parental discipline”. Finally, “parent–school contact” was denoted by the two observed variables “parents initiate contact with teachers” and “teachers initiate contact with parents (parents passively contact teachers)” during the school year, with the values “never = 1, once = 2, two to four times = 3, five or more times = 4”.

The “teacher supervision” dimension used the teacher’s classroom management and attention toward the students and corresponded to question B5, “Do you agree with the following statements about the main course?” Three courses, mathematics, Chinese, and English, were included, each measuring three aspects of the teacher’s attention, questioning, and commendations in the class. A total of nine questions were asked, with a four-point scale ranging from “totally disagree = 1” to “totally agree = 4”. Thus, the three observed variables that make up the latent variable “teacher supervision”, namely, “attention”, “question”, and “commendation”, are the sum of the related questions from the mathematics, Chinese and English teachers at each level.

The peer effect was in the form of two latent variables, “positive peer effect” and “peer misconduct”; question D11 was used to evaluate this dimension. “Did any of the following happen to your best friend?”, including “excellent grades (D11-1), hard-working (D11-2), desire to go to university (D11-3), truancy (D11-4), indiscipline (D11-5), fighting (D11-6), smoking and drinking (D11-7), frequenting internet cafes and game halls (D11-8), early love (D11-9), dropping out of school (D11-10)”. Assignment was on a three-point scale, from “none of them = 1” to “many of them = 3”. After adjustment, we redefined it as follows: The latent variable “positive peer effect” was represented by three observed variables, which were “excellent grades” (D11-1), “hard working” (D11-2), and “purposeful” (D11-3). In addition, the latent variable “peer misconduct” was expressed in terms of the two observed variables “school performance” and “life behavior”. “School performance” was expressed as the sum of questions D11-4, D11-5 and D11-10, forming a variable with values in the range 3–9 (Cronbach’s alpha = 0.77). Similarly, “life behavior” was represented by the sum of questions D11-6, D11-7, D11-8 and D11-9, forming a variable with values in the range 4–12 (Cronbach’s alpha = 0.82).

#### 2.2.3. Confirmatory Factor Analysis

We conducted confirmatory factor analysis (CFA) for the instruments used to measure parental level (parental relationship, parental discipline and parent–school contact), teacher level (teacher supervision), peer level (peer positive behavior and misconduct), and self-level (self pro-sociality and misconduct). The goodness-of-fit index of the parental level was as follows: (1) chi-square/df  =  16.74; (2) RMSEA  =  0.045; (3) GFI  =  0.99; (4) AGFI  =  0.98; (5) TLI  =  0.96; (6) CFI  =  0.98. The goodness-of-fit index of the teacher level was as follows: (1) chi-square/df  =  6.288; (2) RMSEA  =  0.049; (3) GFI  =  1.00; (4) AGFI  =  0.99; (5) TLI  =  0.98; (6) CFI  =  0.99. The goodness-of-fit index of peer level was as follows: (1) chi-square/df  =  23.11; (2) RMSEA  =  0.053; (3) GFI  =  1.00; (4) AGFI  =  0.98; (5) TLI  =  0.99; (6) CFI = 1.00. The goodness-of-fit index of the self level was as follows: (1) chi-square/df  =  12.01; (2) RMSEA  =  0.038; (3) GFI  =  1.00; (4) AGFI  =  0.99; (5) TLI  =  0.99; (6) CFI = 1.00. These outcomes indicate that all the instruments have good validity.

#### 2.2.4. Exogenous Variables

Control variables include gender, residence, being an only child, family financial situation, and parental education level. Age was not included as a control variable, considering that all students in the sample were in grade 8, and their ages did not differ significantly. After reassignment, the dichotomous variables were gender (girl = 0, boy = 1), residence (nonlocal = 0, local = 1) and only child (no = 0, yes = 1). The family’s financial situation was divided into five levels, from “very poor = 1” to “very rich = 5”. Paternal education is converted from education to the corresponding number of years, specifically, “no schooling = 0, elementary school = 6, middle school = 9, special secondary school/technical school = 11, vocational high school/general high school = 12, junior college = 15, undergraduate college = 16, graduate and above = 19”.

### 2.3. Statistical Analysis

This study used Stata version 15.0 for data processing and analysis. First, a descriptive statistical analysis of the variables was performed. Correlation analysis was also carried out for the main variables. Second, the structural equation model (SEM) was used for modeling estimation and fitting.

Specifically, we defined exogenous latent variables at the family level, the teacher level, and the peer level. The endogenous latent variable was the development of self-social behavior. Before constructing the structural equation model, a validation factor analysis was conducted on the latent variables to validate the measurement model, excluding observed variables with factor loadings less than 0.55. Then, path diagrams were constructed by including all latent and control variables following the research needs and relevant hypotheses. After estimating the parameters (maximum likelihood estimate, MLE), direct effects on the development of self-social behavior at the family, teacher, and peer levels and the degree of influence between the variables were derived. Finally, the structural equations were corrected (MI ≥ 3.841) and tested for overall goodness-of-fit. The goodness-of-fit was verified using the chi-square test (χ^2^), root mean square error of approximation (RMSEA), and comparative fit index (CFI).

## 3. Results

### 3.1. Sample Characteristics

A total of 7835 Grade 8 students were included in this study, and the mean age of the adolescents in the sample was 14.52 (SD 0.69). Among them, 3972 male students (50.71%) and 3962 female students (49.29%) were included. As many as 6439 students (82,12%) were from the prefecture in terms of residence distribution, with 1396 students (17.78%) from outside the prefecture occupying the sample. Concerning family structure, the number of “one-child family” was 3563 (45.48%), and the number of multiple-children families was 4272 (54.52%). When the family’s financial situation was considered a continuous variable, the mean was 2.95 (SD 0.60), indicating that most households were in the medium range. After converting parental education level to specific years, the mean value is 10.41 (SD 3.29). The basic sample sociodemographic characteristics (covariates) are illustrated in Table 1.

### 3.2. Correlation Analysis of Central Variables

Table 2 shows the results of the correlation analysis for each central variable. Family effects were used for the latent variables “Parental relationship (F1)”, “Parental discipline (F2)”, and “Parent–school contact (F3)”. These three latent variables contain seven observed variables (x1–x7), among which the correlation coefficients of all variables are less than 0.5, except for the correlation coefficient between parent-initiated school contact and passive contact. This reflected a mutual influence between active contact and passive contact, while the correlations of the other variables were not obvious. The various aspects of the family’s effect on the development of adolescent social behavior are considered. Good family relationships were a facilitator of children’s prosocial development (*p* < 0.01) and were negatively associated with the development of misconduct (*p* < 0.01). The better the relationship between the child and the parents was, the better the development of prosocial behavior.

Similarly, parental discipline showed similar results, with parental discipline being positively related to children’s prosocial development (*p* < 0.01) and negatively related to misconduct (*p* < 0.01), suggesting that the stricter parental discipline was with regard to school, life, and electronics use, the better the child’s prosocial development and the less delinquent behavior (*p* < 0.01). Notably, there was a positive association between parent-initiated teacher contact and children’s prosocial development (*p* < 0.01), and the association with delinquent behavior did not reach statistical significance. However, for children whose teachers-initiated contact with their parents, there was only a positive association with delinquent behavior (*p* < 0.01), and the other associations did not reach statistical significance. The family plays an essential role in the behavioral development of adolescents.

At the teacher level, the correlation coefficients between the three observed variables, teacher attention, teacher questions, and teacher commendations all exceeded 0.5, suggesting that changes in any one dimension may cause changes in other dimensions. The data also showed that teacher attention, questioning, and commendations all positively affected adolescent prosocial development (*p* < 0.01). Simultaneously, these three areas were negatively associated with adolescent misconduct (*p* < 0.01). They demonstrated that teachers also play an essential role in the development of adolescent behavior.

Even more significant is the impact of peer behavior on the development of adolescent social behavior. The peer effect consisted of two latent variables, positive peer behavior and peer misconduct. The three observed variables represented positive peer behavior with correlation coefficients greater than 0.5, and the two observed variables represented negative peer behavior with correlation coefficients greater than 0.5. At the same time, positive peer behavior was positively correlated with adolescents’ development of pro-sociality (*p* < 0.01) and negatively correlated with the development of misconduct (*p* < 0.01). Positive peer behavior was positively correlated with adolescents’ development of pro-sociality (*p* < 0.01) and negatively correlated with the development of misconduct (*p* < 0.01). Conversely, peer misconduct was negatively correlated with adolescent prosocial development (*p* < 0.01) and positively correlated with misconduct (*p* < 0.01). It also indicated that peer groups influence adolescent behavioral development, and it is necessary to pay close attention to group behavior to prevent and reduce destructive behaviors.

### 3.3. Structural Equation Modeling Analysis

Before constructing the model, factor analysis was conducted to explore the validity of identified factors and the relationship of observed variables with latent variables. After adjustment, the results showed that the factor loadings of all the observed variables at three different levels were above 0.55 (Table 1), revealing that the observed variables used in the study could explain the latent variables separately and objectively reflect the significance of the latent variables.

After constructing the structural equation model (Figure 1), we estimated the model using the maximum likelihood method, followed by a correction (MI ≥ 3.841), and finally tested the overall model for the goodness of fit. The results showed that χ²/df = 16.11, *p* < 0.001, RMSEA = 0.044, CFI = 0.934. The chi-square values, although significant, are likely to produce model mismatch results given that they are highly susceptible to distortion with large samples. Thus, two other indicators were calculated (RMSEA and CFI), with the RMSEA value being less than 0.05, an acceptable result, and the CFI exceeding the critical value of 0.90, giving an overall good fit to the model.

Concerning the relationship of the independent variables with the dependent variables (Table 3 and Figure 2), the coefficients of determination (*R^2^*) of the independent variables on the two dependent variables were 0.483 and 0.397, respectively, which reached a moderate level. The active effects on adolescent prosocial development were parental relationship (β = 0.128, *p* < 0.01), parental discipline (β = 0.108, *p* < 0.01), teacher supervision (β = 0.078, *p* < 0.01), and positive peer behavior (β = 0.228, *p* < 0.01). Peer misconduct is detrimental to adolescent prosocial development (β = −0.114, *p* < 0.01). The impact of parent-school contact in this area did not reach a significant level. With regard to adolescent misconduct, parental relationship (β = −0.085, *p* < 0.01), parental discipline (β = −0.122, *p* < 0.01), teacher supervision (β = −0.015, *p* < 0.01), and peer-positive behavior (β = −0.131, *p* < 0.01) significantly reduced the occurrence of such events. Parent–school contact (β = 0.057, *p* < 0.01) and peer misconduct (β = 0.458, *p* < 0.01) were positively associated with adolescent misconduct. As a result, good family relationships, parental and teacher supervision in daily life and learning, combined with good peer behavior, can effectively reduce adolescent misconduct and help adolescents develop good behavior and healthy personalities. A peer group’s delinquent behavior exacerbates the tendency for adolescents to behave inappropriately, and too much contact with undesirable peers is detrimental to the development of prosocial behavior. It can even lead to more severe undesirable behavior.

This study’s covariates were gender, residence, being an only child, family financial situation, and parental education level. Specifically, boys were less prosocial than girls (β = −0.055, *p* < 0.01) and more likely to exhibit delinquent behavior (β = 0.108, *p* < 0.01). Adolescents whose permanent residence was local were less prosocial than nonlocal adolescents (β = −0.039, *p* < 0.01), and the difference in delinquent behavior was not significant. Only children also outperformed multiples in terms of behavioral development (β = 0.035, *p* < 0.05) and had a lower incidence of delinquent behavior (β = −0.038, *p* < 0.01). The better the family’s economic conditions are, the better the children’s social performance in terms of pro-sociality (β = 0.028, *p* < 0.05). Finally, the level of paternal education also significantly promoted children’s social behavior (β = 0.083 *p* < 0.01) and reduced delinquent behavior (β = −0.033, *p* < 0.01).

## 4. Discussion

Our study investigated the influences of significant others on adolescent social behaviors by looking at three aspects: parents, teachers, and peers. The results show that parental relationships, family–school contact, parental discipline, teacher supervision, and peer group behavior all have direct and significant impacts on adolescent behavior development, suggesting that significant others play different roles in the development of adolescent behavior.

The study results showed that good parent—child relationships significantly improved children’s behavioral development and reduce the incidence of delinquent behavior. This finding is similar to the results of other studies. A good family relationship creates a warm, stable communication channel that allows parents to communicate openly with their children, which subconsciously influences adolescents’ personalities and behavior [11,32]. Simultaneously, as parental supervision increased, the level of delinquent behavior decreased. This is explained by sociological control theory, which states that people are born with a tendency to commit crimes or engage in delinquency and are prone to do so when they lack control [33]. Therefore, parental supervision can play a controlling role in the behavior of adolescents [27,34,35]. The intrinsic association between parental supervision and parent—child relationships is positive, which is in line with the findings of Keijsers [36]. Through supervision, parents and children have more opportunities to interact with each other; the relationship will become more robust, and emotional attachment can grow closer. With timely communication, parents can transmit positive experiences and thoughts to their children, reducing misconceptions and cognitive deviations affecting their children’s behavior [37]. However, this differs from the findings of Fletcher et al., who argue that strict parental supervision can even cause rebellious behavior in adolescents. In this regard, we speculate that different cultural backgrounds contribute to this difference. Children in Western societies are socialized to grow up independently, and parents need only to provide appropriate guidance; too much intervention can deprive children of their free play and intensify parent—child conflicts, thus inducing undesirable behavior [38]. On the other hand, Chinese families are influenced by traditional Confucianism, where parental supervision is a responsibility and an obligation to their children. Through discipline, parents can reduce tendencies in their children’s behavior that diverge from social and cultural morals [33]. However, it is worth noting that this study found that home–school contact does not impact the prosocial behavior of adolescents. Instead, it exacerbates the occurrence of their delinquent behavior. A plausible explanation is that passive contact plays an essential role in this mechanism; parents’ frequent passive contact with teachers tends to mean that such children appear to have more delinquent behavior themselves. Parents play the role of initiating mentors in the development of their children. Therefore, parents need to build a harmonious and stable family atmosphere, provide help and counseling to their children, and communicate and correct their children when delinquent behavior occurs to ensure their children’s healthy development.

We found as teacher supervision increased, the better the adolescent performed at prosocial behavior as measured by teacher influence, with a corresponding decrease in delinquent behavior. One of the groups with which adolescents are most in contact, teachers, also have a crucial influence on them. Teacher discipline often complements parental supervision, and when adolescents are out of the reach of their parents during school hours, teachers take over this role and correct adolescent behavioral development [26]. This result is also consistent with the findings of existing research. By showing care and attention to students, teachers can also build their emotional attachment to each other, enabling them to be more receptive to the teacher’s advice and opinions and develop good behavioral habits. Simultaneously, teachers’ questions and praise in the classroom can motivate students and increase their self-confidence in school and life [24,39]. Although some foreign scholars argue that traditional education is not universally applicable [40], Chinese educational models and cultural traditions suggest that teachers have a particular authoritative role in student education and development and act as ‘guides’ for young people’s development. With this in mind, teachers must be as attentive as possible to their students’ lives and psychology, providing them with help and guidance and regulating their behavior and psychological development while ensuring that they complete their learning tasks.

The findings further confirmed that there was also a significant role in the influence of peer behavior on adolescent behavioral development. Prosocial behavior in peer groups promoted acceptable behavioral norms in oneself, while poor behavior in peers led to delinquency in oneself. An explanation for this can be found in differential interaction theory. People learn and imitate their peer group’s behavior and develop similar values to the group to fit better with the group [28]. The findings of this study are also consistent with the results of existing research. Youth have the most contact with their peers, both in and out of school, and receive warmth and support in their groups. Forming friends with positive energy can significantly reduce the emergence of delinquent behavior [41]. However, some minority groups need special attention. These young people are often very rebellious, seek extreme individuality, and have difficulty fitting in with the mainstream culture, so they congregate to form minority groups. They are at a high risk of delinquent behavior [28]. Therefore, while encouraging young people to integrate into social groups, it is also vital to strengthen guidance and education to reduce small groups with fringe behavior.

Differences in the behavioral development of adolescents are also reflected in several other areas. Boys are more likely to exhibit delinquent behavior than girls. Scholars generally believe that this is related to biological structure and the state of psychological maturity. Girls mature earlier than boys and tend to deal with problems in a rational way and behave in a more self-disciplined manner. Boys, on the other hand, are more likely to adopt impulsive behavior [42]. In terms of other family factors, being an only child, having a privileged economic family background, and having parents with a high level of education are also beneficial to adolescent behavioral development. In these cases, parents have high expectations for their children’s academic achievements, invest a lot of time and money in their children’s development and are more competent in disciplining them [43].

Although this study yielded a large number of significant findings, several limitations remain. Firstly, although the CEPS data used in this study are authoritative, it is not a questionnaire that professionally measures adolescent behavioral delinquency and falls short in terms of professionalism and relevance. Secondly, the latest available data from the CEPS survey are still from the 2014–2015 follow-up period. With the changes in social life and the economy, the continuation of the global epidemic will bring more challenges to the psychological and behavioral development of adolescents, so a new wave of investigation is necessary. Thirdly, the mechanism of action explored in this study is the direct influence of significant others on adolescent behavioral development, but the indirect factors are challenging to identify and exclude. Future research could be based on data surveys specifically targeting adolescent behavioral problems, while more complicated and pathways of action need to be established, and more comprehensive models need to be developed.

## 5. Conclusions and Implications

Using mainly CEPS data, this study aims to analyze the factors influencing adolescent behavior development and the mechanisms at play. The study examines the influence of family, teachers, and peers on adolescents’ prosocial and delinquent behaviors. It is intended to provide a theoretical basis for the development of theory, the formulation of policy, and the improvement of social work. This study’s findings have theoretical and practical implications, and the results enrich the empirical research for this age cohort in China. The results can also positively improve systems and measures to regulate the behavior of young people at different levels. It is necessary to pay attention to the values of significant others in the process of adolescent development to provide a healthy, harmonious, and stable environment for adolescents to grow up in.

## Figures and Tables

**Figure 1 brainsci-13-00191-f001:**
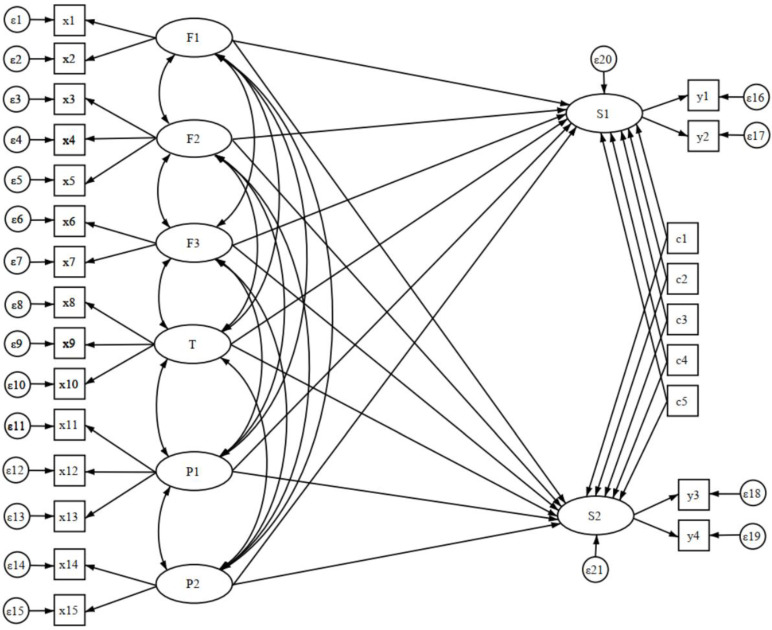
Structural equation model path diagram.

**Figure 2 brainsci-13-00191-f002:**
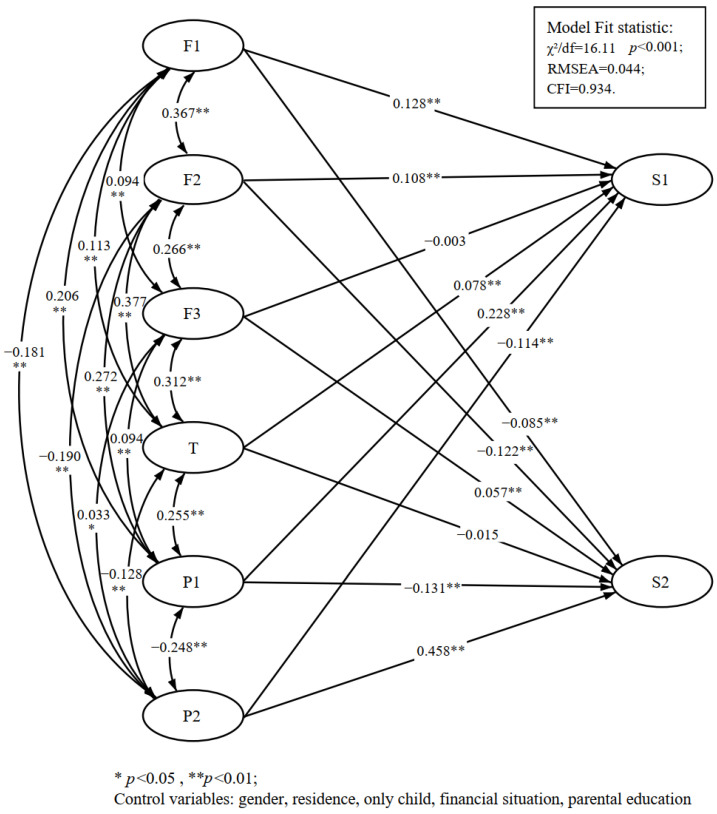
Structural equation model normalization results.

**Table 1 brainsci-13-00191-t001:** Definition and description of variables, *n* = 7835.

Latent Variables	Observation Variables	Value Range	Mean (Std.) or No. (%)	Factor Loading
Parental relationship (F1)	with father (x1)	[1, 3]	2.51 (0.58)	0.64
	with mother (x2)	[1, 3]	2.71 (0.50)	0.68
Parental discipline (F2)	study (x3)	[2, 6]	4.73 (0.98)	0.70
	life (x4)	[2, 6]	4.15 (1.09)	0.56
	electronics (x5)	[2, 6]	4.78 (1.12)	0.57
Parent–school contact (F3)	initiative (x6)	[1, 4]	2.33 (1.01)	0.87
	passive (x7)	[1, 4]	2.09 (0.98)	0.59
Teacher supervision (T)	attention (x8)	[3, 12]	8.25 (2.25)	0.77
	question (x9)	[3, 12]	7.65 (2.35)	0.90
	commendation (x10)	[3, 12]	7.18 (2.55)	0.68
Peer positive behavior (P1)	excellent grades (x11)	[1, 3]	2.36 (0.60)	0.82
	hard-working (x12)	[1, 3]	2.38 (0.62)	0.89
	purposeful (x13)	[1, 3]	2.62 (0.58)	0.68
Peer misconduct (P2)	school performance (x14)	[3, 9]	3.34 (0.90)	0.77
	life behavior (x15)	[4, 12]	4.79 (1.46)	0.96
Pro-sociality (S1)	mannered (y1)	[1, 5]	4.07 (0.97)	0.77
	friendly (y2)	[1, 5]	4.21 (0.85)	0.71
Misconduct (S2)	self-behavior (y3)	[6, 30]	7.29 (2.38)	0.91
	language and psychology (y4)	[4, 20]	8.10 (2.93)	0.57
Covariates (C)	gender	girl = 0	3862 (49.29)	-
		boy = 1	3972 (50.71)	-
	residence	nonlocal = 0	1396 (17.82)	-
		local = 1	6439 (82.18)	-
	only child	no = 0	4272 (54.52)	-
		yes = 1	3563 (45.48)	-
	financial situation	[1, 5]	2.95 (0.60)	-
	parental education	[0, 19]	10.41 (3.29)	-

**Table 2 brainsci-13-00191-t002:** Correlation analysis of the main variables *n* = 7835.

Variables	x1	x2	x3	x4	x5	x6	x7	x8	x9	x10	x11	x12	x13	x14	x15	y1	y2	y3	y4
**x1**	1.00																		
**x2**	0.44 *	1.00																	
**x3**	0.20 *	0.20 *	1.00																
**x4**	0.10 *	0.12 *	0.38 *	1.00															
**x5**	0.11 *	0.13 *	0.39 *	0.37 *	1.00														
**x6**	0.07 *	0.07 *	0.21 *	0.10 *	0.09 *	1.00													
**x7**	0.04 *	0.01	0.13 *	0.06 *	0.02	0.51 *	1.00												
**x8**	0.16 *	0.16 *	0.24 *	0.16 *	0.14 *	0.14 *	0.13 *	1.00											
**x9**	0.18 *	0.17 *	0.26 *	0.17 *	0.17 *	0.14 *	0.11 *	0.70 *	1.00										
**x10**	0.20 *	0.18 *	0.21 *	0.12 *	0.10 *	0.12 *	0.09 *	0.51 *	0.62 *	1.00									
**x11**	0.09 *	0.11 *	0.13 *	0.12 *	0.11 *	0.08 *	0.02 *	0.16 *	0.17 *	0.16 *	1.00								
**x12**	0.11 *	0.13 *	0.17 *	0.14 *	0.15 *	0.08 *	0.01 *	0.18 *	0.23 *	0.18 *	0.74 *	1.00							
**x13**	0.10 *	0.14 *	0.15 *	0.12 *	0.15 *	0.07 *	−0.01	0.13 *	0.16 *	0.12 *	0.56 *	0.60 *	1.00						
**x14**	−0.08 *	−0.10 *	−0.07 *	−0.05 *	−0.12 *	0.01 *	0.05 *	−0.07 *	−0.09 *	−0.05 *	−0.15 *	−0.18 *	−0.20 *	1.00					
**x15**	−0.10 *	−0.13 *	−0.10 *	−0.09 *	−0.17 *	0.02 *	0.06 *	−0.09 *	−0.12 *	−0.07 *	−0.17 *	−0.21 *	−0.21 *	0.74 *	1.00				
**y1**	0.11 *	0.13 *	0.15 *	0.11 *	0.11 *	0.06 *	0.01	0.15 *	0.15 *	0.13 *	0.20 *	0.24 *	0.25 *	−0.18 *	−0.20 *	1.00			
**y2**	0.13 *	0.16 *	0.16 *	0.09 *	0.09 *	0.05 *	0.01	0.17 *	0.15 *	0.14 *	0.19 *	0.23 *	0.23 *	−0.13 *	−0.13 *	0.55 *	1.00		
**y3**	−0.12 *	−0.14 *	−0.15 *	−0.11 *	−0.18 *	0.01	0.08 *	−0.10 *	−0.14 *	−0.10 *	−0.21 *	−0.26 *	−0.25 *	0.39 *	0.49 *	−0.27 *	−0.20 *	1.00	
**y4**	−0.18 *	−0.15 *	−0.15 *	−0.10 *	−0.15 *	−0.02	0.03	−0.11 *	−0.17 *	−0.21 *	−0.16 *	−0.21 *	−0.11 *	0.19 *	0.29 *	−0.20 *	−0.18 *	0.53 *	1.00

* *p* < 0.01 x1: relationship with father, x2: relationship with mother, x3: parental discipline (study), x4: parental discipline (life), x5: parental discipline (electronics), x6: parent–school contact (initiative), x7: parent–school contact (passive), x8: teacher supervision (attention), x9: teacher supervision (question), x10: teacher supervision (commendation), x11: peer positive behavior (excellent grades), x12: peer positive behavior (hard-working), x13: peer positive behavior (purposeful), x14: peer misconduct (school performance), x15: peer misconduct (life behavior), y1: pro-sociality (mannered), y2: pro-sociality (friendly), y3: misconduct (behavior), y4: misconduct (language and psychology).

**Table 3 brainsci-13-00191-t003:** Structural model path coefficients *n* = 7835.

Pathways	*β*	*S.E.*	*p*	*R* ^2^
**Pro-sociality (S1)**				
←Parental relationship (F1)	0.128	0.022	<0.001	0.483
←Parental discipline (F2)	0.108	0.022	<0.001	
←Parent-school contact (F3)	−0.003	0.016	0.863	
←Teacher supervision (T)	0.078	0.017	<0.001	
←Peer positive behavior (P1)	0.228	0.018	<0.001	
←Peer misconduct (P2)	−0.114	0.021	<0.001	
←Gender	−0.055	0.014	<0.001	
←Residence	−0.039	0.013	0.002	
←Only child	0.035	0.014	0.011	
←Financial situation	0.028	0.015	0.033	
←Parental education	0.083	0.014	<0.001	
**Misconduct (S2)**				
←Parental relationship (F1)	−0.085	0.020	<0.001	0.397
←Parental discipline (F2)	−0.122	0.021	<0.001	
←Parent-school contact (F3)	0.057	0.016	<0.001	
←Teacher supervision (T)	−0.015	0.015	0.258	
←Peer positive behavior (P1)	−0.131	0.018	<0.001	
←Peer misconduct (P2)	0.458	0.024	<0.001	
←Gender	0.108	0.011	<0.001	
←Residence	−0.005	0.011	0.661	
←Only child	−0.038	0.014	0.001	
←Financial situation	−0.000	0.014	0.976	
←Parental education	−0.033	0.013	0.005	

*β* = Standardized path coefficients, *S.E*. = Robust Standard error, *p* = *p* value.

## Data Availability

The datasets used and/or analyzed during the current study are available from the corresponding author on reasonable request.

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
