# Peer review of "Parental, Teacher and Peer Effects on the Social Behaviors of Chinese Adolescents: A Structural Equation Modeling Analysis"

_brainsci, 2023, doi:10.3390/brainsci13020191_

Round 1
Reviewer 1 Report
The study has potential, but as currently written and described contains numerous limitations. Addressing issues bellow would strengten manuscript.
Introduction:
The opening paragraph should concisely describe what the reader should expect to read about in the manuscript and know exactly what the Authors are examining. However, these paragraphs are very vague and do not inform readers about what their research question or questions will be.
Subsequent sections need further development. Statements are made with no elaboration to support Authors’ claims. Statements such as “As seen, the family's role in adolescent behavior development varies across cultures and still needs further validation” are made but background information about these cultural differences are absent. Authors mention “poor family relationships” but never describe what this means. This pattern is seen in multiple places in the introduction. Authors propose the presence of conflicts in the literature but don’t sufficiently develop or support these thoughts. What are the conflicting models and/or findings? If indirect effects are mentioned, what evidence (theoretical, methodological, etc.) justify this assertation? Greater elaboration is needed throughout.
The intro is also devoid of any theory. No hypothesis is provided. Hence, the study comes across as exploratory. It lacks a sound rational and clarity.
Methods:
Measures are not clearly described. What are the psychometric properties for these measures? If none exist, an EFA and then CFA need to be performed before any discussion of a model. Authors need to provide evidence that the measures used tap into the construct of interest. Furthermore, this is a multi-level study. I did not see mentions of ICC or clustering being accounted for. Because of these methodological and statistical issues, further assessment of the manuscript cannot be made.
Author Response
Dear Editors and Reviewers,
Thanks very much for taking your time to review this manuscript. We really appreciate all your generous comments and suggestions! Please find my itemized responses in below and my revisions in the re-submitted files.
Response 1: The section on the research question is formulated in our 1.3 Research Interest. Our consideration was to present our research question and hypothesis after the literature review and therefore placed this section in the final part of the introduction.
1.3 Research Interest
The development of adolescent behavior has received increasing interest in the literature, and a large number of a priori findings have laid the foundation for future theoretical and practical products. However, there is still room left for improvement. Most studies have taken a single perspective on the influence of adolescent delinquency, with the independent and dependent variables in only one dimension. However, the human environment is complex and interactive, with parents, teachers, and peers influencing adolescent values and behaviors differently. There are still controversies and gaps in previous research, with scholars not reaching a consensus on parental and teacher supervision and insufficient research on whether parent–school contact im-pacts adolescent behavioral development. Whether the mechanisms of influence of significant others on adolescents remain reliable remains to be studied.
Thus, in this paper, parents, teachers, and peers were all considered as exogenous variables, and the endogenous variables included both positive and negative aspects of adolescent behavioral development. Structural equation modeling was used to investigate the underlying mechanisms.
Response 2: Some statements in the introduction have been revised. Appropriate additions have been made to the literature review, and our hypotheses have been added after each dimension. The main modifications were made in section 1.2Family, Teacher, Peer Effects.
Response 3: We conducted confirmatory factor analysis (CFA), displayed in section 2.2.3:
2.2.3. Confirmatory factor analysis
We conducted confirmatory factor analysis (CFA) for the instruments used to measure parental level (parental relationship, parental discipline and parent–school contact), teacher level (teacher supervision), peer level (peer positive behavior and misconduct), and self level (self pro-sociality and misconduct). The goodness-of-fit index of parental level was as follows: (1) chi-square/df = 16.74; (2) RMSEA = 0.045; (3) GFI = 0.99; (4) AGFI = 0.98; (5) TLI = 0.96; (6) CFI = 0.98. The goodness-of-fit index of teacher level was as follows: (1) chi-square/df = 6.288; (2) RMSEA = 0.049; (3) GFI = 1.00; (4) AGFI = 0.99; (5) TLI = 0.98; (6) CFI = 0.99. The goodness-of-fit index of peer level was as follows: (1) chi-square/df = 23.11; (2) RMSEA = 0.053; (3) GFI = 1.00; (4) AGFI = 0.98; (5) TLI = 0.99; (6) CFI = 1.00. The goodness-of-fit index of self level was as follows: (1) chi-square/df = 12.01; (2) RMSEA = 0.038; (3) GFI = 1.00; (4) AGFI = 0.99; (5) TLI = 0.99; (6) CFI = 1.00. These outcomes indicate that all the instruments have good validity.
Response 4: With regard to your question about multi-level studies, I am sorry that we did not fully consider it before. However, after reviewing, we did not find enough relevant studies that could guide us in this type of analysis, so we would like to ask you if you could provide us with some studies or books that we could study and we would be happy to make the next step in our revision.
We would like to thank the referee again for taking the time to review our manuscript. And we wish you a happy New Year!
Reviewer 2 Report
This paper analyses data on a large (N=7835) cohort of Chinese adolescents, focusing on prosocial behaviours as a dependent variable, analysing the interplay of reported behaviours and attitudes of parents, teachers and peers as predictor variables, as well as certain demographic factors, which might influence the reported outcome measures, in terms of prosocial behaviours. The data are analysed using structural equational modelling, which is an ideal approach to understanding complex interactions of predictor variables in a large data set.
Results are interesting and important and have implications both for theory development, and professional practice in teaching and student support. These results deserve replication in different cultures, and do make a significant contribution to knowledge in this field. The paper is very well written, with a very clear presentation of findings, and a discussion of results.
Author Response
Dear Editors and Reviewers,Thanks very much for taking your time to review this manuscript. We are very pleased with your comments on our articles and hope that you will contact us if you have any questions to discuss.
We have made some revisions to our manuscript :
- We conducted confirmatory factor analysis (CFA), displayed in section 2.2.3:
- We added to Table 3 a description of the coefficient of determination (R2) of the independent variable on the two dependent variables (individual pro-sociality and deviant behavior) were 0.483 and 0.397 respectively, which reached a moderate level.
- Some statements in the introduction have been revised. Appropriate additions have been made to the literature review, and our hypotheses have been added after each dimension. The main modifications were made in section 1.2Family, Teacher, Peer Effects.
We would like to thank the referee again for taking the time to review our manuscript. And we wish you a happy New Year!
Reviewer 3 Report
Dear authors:
First of all, I congratulate you on your very interesting study on the influence of parents, teachers and peers on the prosocial or delinquent behaviour of adolescents in school.
However, it is necessary to improve some aspects that have not been well presented and present doubts:
1. I see the need to include some reliability and validity indices such as Mcdonalds' Omega, Composite Reliability and Average Variance Extracted (AVE) to the scales or instruments used, in addition to Cronbach's Alpha.
2. I cannot find the explained variance or coefficient of determination of the structural equation model proposed, and see when it explains this model.
Author Response
Dear Editors and Reviewers,
Thanks very much for taking your time to review this manuscript. We really appreciate all your generous comments and suggestions! Please find my itemized responses in below and my revisions in the re-submitted files.
Response 1: The scale we used in this study was taken from the China Education Panel Survey, which was validated by a pre-survey reliability test. At the same time, we listed the standardized factor loadings in Table1 (all >0.5 required), and to further validate, we conducted confirmatory factor analysis (CFA), displayed in section 2.2.3:
2.2.3. Confirmatory factor analysis
We conducted confirmatory factor analysis (CFA) for the instruments used to measure parental level (parental relationship, parental discipline and parent–school contact), teacher level (teacher supervision), peer level (peer positive behavior and misconduct), and self level (self pro-sociality and misconduct). The goodness-of-fit index of parental level was as follows: (1) chi-square/df = 16.74; (2) RMSEA = 0.045; (3) GFI = 0.99; (4) AGFI = 0.98; (5) TLI = 0.96; (6) CFI = 0.98. The goodness-of-fit index of teacher level was as follows: (1) chi-square/df = 6.288; (2) RMSEA = 0.049; (3) GFI = 1.00; (4) AGFI = 0.99; (5) TLI = 0.98; (6) CFI = 0.99. The goodness-of-fit index of peer level was as follows: (1) chi-square/df = 23.11; (2) RMSEA = 0.053; (3) GFI = 1.00; (4) AGFI = 0.98; (5) TLI = 0.99; (6) CFI = 1.00. The goodness-of-fit index of self level was as follows: (1) chi-square/df = 12.01; (2) RMSEA = 0.038; (3) GFI = 1.00; (4) AGFI = 0.99; (5) TLI = 0.99; (6) CFI = 1.00. These outcomes indicate that all the instruments have good validity.
Response 2: We added to Table 3 a description of the coefficient of determination (R2) of the independent variable on the two dependent variables (individual pro-sociality and deviant behavior) were 0.483 and 0.397 respectively, which reached a moderate level.
Response 3: Some statements in the introduction have been revised. Appropriate additions have been made to the literature review, and our hypotheses have been added after each dimension. The main modifications were made in section 1.2Family, Teacher, Peer Effects.
We would like to thank the referee again for taking the time to review our manuscript. And we wish you a happy New Year!